Surface-functionalized PAN fiber membranes for the sensitive detection of airborne specific markers

Varvarovska Leontyna 1
Sopko Bruno bruno.sopko@gmail.com 2 3
Gaskova Dana 4
Bartl Tomas 4
Amler Evzen 5
Jarosikova Tatana 1
1 Department of Natural Sciences, Faculty of Biomedical Engineering, Czech Technical University in Prague , Kladno , Czech Republic
2 Department of Medical Chemistry and Biomedical Biochemistry, Second Faculty of Medicine and Faculty Hospital Motol, Charles University Prague , Prague , Czech Republic
3 Laboratory of Advanced Biomaterials, University Centre for Energy Efficient Buildings, Czech Technical University in Prague , Bustehrad , Czech Republic
4 Institute of Physics of Charles University, Faculty of Mathematics and Physics, Charles University Prague , Prague , Czech Republic
5 Department of Biophysics, Second Faculty of Medicine, Charles University Prague , Prague , Czech Republic
Franco Bernardo
Electronic publication date: 2024 Oct 23
Publication date: 2024
Volume: 12
Electronic Location ID: e18077
Received 2023 Nov 7; Accepted 2024 Aug 20
Copyright: ©2024 Varvarovska et al.
Copyright year: 2024
Copyright holder: Varvarovska et al.
License: This is an open access article distributed under the terms of the Creative Commons Attribution License, which permits unrestricted use, distribution, reproduction and adaptation in any medium and for any purpose provided that it is properly attributed. For attribution, the original author(s), title, publication source (PeerJ) and either DOI or URL of the article must be cited.
License URL: https://creativecommons.org/licenses/by/4.0/

Keywords: Functionalized PAN nanofibers, Antibody-based nanobiosensor, Nanofiber biosensors, Air detection, Airborne pathogen detection

Funding: Czech Technical University in Prague through the Student Grant Competition of Czech Technical University SGS22/199/OHK4/3T/17 The research was financially supported by Czech Technical University in Prague through the Student Grant Competition of Czech Technical University (SGS22/199/OHK4/3T/17).

==============================
PAN fibers are characterized by having a large surface-to-volume ratio and small pores, which are beneficial for applications in filtration and specific molecular detection systems. Naturally, larger items are filtered, and a lower ratio between specific and nonspecific binding is expected since small pores do not allow larger elements to penetrate through membranes; thus, nonspecific binding is enhanced. We prepared and tested fiber membranes (diameter cca 700 nm) functionalized with a specific antibody to prove that even microscopic systems such as bacteria could be specifically identified. In addition, we established a methodology that enabled the effective binding of bacteria in not only an aqueous environment but also air. Our data clearly prove that even large systems such as bacteria could be specifically identified by fiber membranes surface-functionalized with a specific antibody. This research opens the door to the construction of biosensors for the fast, inexpensive, and sensitive identification of airborne bacterial contaminants and other airborne pollutants.

Introduction

Currently, wastewater (Mao et al., 2020; Mao et al., 2021; Khodayari et al., 2021; Markosian & Mirzoyan, 2019) and air pollution monitoring are priceless techniques enabling the early detection of pathogens and toxic substances, such as heavy metals, drugs, or pesticides, and successful diagnostics (Eltzov et al., 2011; Ghasemi et al., 2022; Ranjbar & Shahrokhian, 2018; Rajamanickam & Yoon Lee, 2022; Willner & Vikesland, 2018; Su et al., 2012; Thakur & Kumar, 2022; Ventura et al., 2020). Due to the events associated with COVID-19, new detection and monitoring methods that could prevent such widespread occurrences are still being developed (Mao et al., 2020). However, in many cases, conventional methods are still used to detect the abovementioned toxic and harmful substances. For pathogen detection and characterization, techniques that are considered as the gold standard, such as culture-based methods, polymerase chain reaction (PCR), and enzyme-linked immunosorbent assay (ELISA), are utilized. Although these methods show high sensitivity and precision, they have several significant disadvantages—for example, the majority of those need to transfer samples to specialized laboratories where the detection is carried out. In addition to being time-consuming processes, these methods also require specialized operators and expensive equipment (Eltzov et al., 2011; Ghasemi et al., 2022; Ranjbar & Shahrokhian, 2018; Rajamanickam & Yoon Lee, 2022). However, newly created sensors and biosensors can be used to successfully solve these problems (Scheller et al., 2001; Byrne et al., 2009).

Fast and sensitive environmental air pollution control is crucial for early diagnostics and disease prevention. Several studies have focused on developing novel methods for detecting and monitoring indoor and outdoor pollutants, such as fine particulate matter (PM2.5) (Eltzov et al., 2011; Kim et al., 2018; Shuvo et al., 2020; Deng et al., 2021; Matulevicius et al., 2016). For the effective detection of PM2.5, one of the critical factors in lung cancer (Kim et al., 2018), the real-time monitoring of the indoor environment and changes within this environment has been studied using the bioluminescence of bacterial reporters (Eltzov et al., 2011). The substances monitored by the afore mentioned methods materials can be captured and detected without any additional difficulties. Currently, the detection of airborne pathogens is more challenging. Many sensitive and rapid techniques have been developed for detecting and quantifying pathogens and other substances in different, but always aqueous, matrices (water, body fluids, food, soil, etc.) (Rajamanickam & Yoon Lee, 2022; Ventura et al., 2020; Al-Taie et al., 2020; Ménard-Moyon, Bianco & Kalantar-Zadeh, 2020; Atik et al., 2023; Sarabaegi, Roushani & Hosseini, 2021). However, the detection of airborne pathogens directly from the air is problematic. Most current airborne pathogen detection methods require postcollection processing (such as the storage, cleaning, and enrichment of samples) in addition to sampling (Ghasemi et al., 2022; Rajamanickam & Yoon Lee, 2022; Al-Taie et al., 2020; Bhardwaj et al., 2021; Yang et al., 2020; Prieto-Simón et al., 2015; Liu et al., 2018; Triadó-Margarit, Cáliz & Casamayor, 2022). Even biosensors created for this purpose face problems with regard to maintaining a proper environment for the biosensing layer and the electrode (Ghasemi et al., 2022; Al-Taie et al., 2020; Bhardwaj et al., 2021; Yang et al., 2020; Liu et al., 2018). For instance, the detection of the bacteria Streptococcus agalactiae, the airborne bacteria that causes pneumonia and meningitis, can be achieved by a nanofiber fluorescent immunosensor. However, antibodies immobilized on the nanomaterials involved in the detection method require the use of liquid samples (broth medium, water, buffer, etc.) to maintain a sufficiently humid environment (Ghasemi et al., 2022; Prieto-Simón et al., 2015). Other pathogens, such as the facultative pathogen bacterium Escherichia coli or the virus SARS-CoV-2, are also airborne pathogens that must be detected using liquid samples. The possibility of detecting these pathogens directly from the air is difficult due to the extremely specific conditions needed for the antibody used as the bioreceptor (Yasri & Wiwanitkit, 2022; Petrovszki et al., 2021).

The biological activity of antibodies depends on the surrounding conditions. In addition to the temperature, pH, and electrostatic repulsion, a sufficient amount of water is essential to carry out the antibody-antigen reaction (Armstrong, 2008; Reverberi & Reverberi, 2007; Wang et al., 2012). Aqueous media allows the stability of the antibody structure to be maintained through hydrogen bonding (Slocik et al., 2021). Moreover, hydrogen bonds facilitate weak interactions that play a key role in the antibody-antigen reaction. Without liquid media or at least sufficient humidity, the antibody would lose its structure and the ability to recognize and bind antigens (Byrne et al., 2009; Slocik et al., 2021; Van Oss, Good & Chaudhury, 1986; Janeway Jr et al., 2001; Holford, Davis & Higson, 2012; Borrebaeck, 2000).

Herein we have developed a novel system, capable of detecting air borne pathogens, and overcoming the above-mentioned problems of antigen-antibody interaction in the air systems. The PAN fibers functionalized with anti-E. coli antibodies served as the detector, and together with the newly developed air filtration system with nebulizer ensured sufficient amount of water to for the antigen-antibody reaction in comparison with the “dry” air.

Materials & Methods

Material

The PAN polymer (average Mw 150,000) used for fiber fabrication was purchased from Sigma-Aldrich (St. Louis, MO, USA), product No. 181315. The prepared fibers were functionalized by antibody immobilization. Rabbit polyclonal IgG anti-E. coli antibodies (4329-4906) were purchased from Bio-Rad (Hercules, CA, USA).

E. coli, a Gram-negative bacterium, was used as a model organism for detection (Zendehel, Goli & Hajibabaei, 2019; Hu et al., 2017). The reference bacterial strains were provided by the bacterial collection of the University of Chemistry and Technology, Prague. The cultivation of bacterial colonies was performed on agar medium. Chemicals used for the preparation of the agar medium (NaCl, peptone, agar, and yeast extract) were obtained from Sigma-Aldrich.

Fiber fabrication

For the detection of bacteria, PAN fibers were supplied by Syndat, Czech Republic. According to the information from Syntadfor fabrication of fine fibers, PAN was prepared from a mixture of PAN polymer and N, N-dimethylformamide (DMF). The fiber membrane (17.3 ± 0.6 g.m−2) was then fabricated by an electrospinning process by Syndat. (Bayrak, 2022; Mercante et al., 2021; Halicka & Cabaj, 2021; Senthil et al., 2022; Ramakrishna et al., 2005; Lin, 2011; Brown & Stevens, 2007).

Afterward, the visualization and characterization of the fabricated fibers were performed by using a scanning electron microscope (Vega3 SB; Tescan, Czech Republic) (Bayrak, 2022). For observation through SEM, samples of PAN fibers were collected (Sputter Coater Q150R; Quorum Technologies, East Sussex, UK). Anti-E. coli PAN fibers (functionalized NFs) needed to be stored in the buffer media. Thus, to visualize such fibers through SEM, it was necessary to lyophilize the samples. The lyophilization of functionalized fibers was performed in a vacuum at −75 °C for 1 h (VirTis Benchtop Pro Freeze Dryer; SP Scientific, Warminster, PA, USA). After the elimination of liquid in the sample, visualization was performed as described above.

Functionalization of PAN fibers

After the preparation of PAN fibers, the surface modification was necessary. Surface modification—reduction—was performed to obtain suitable functional groups (−NH3) chemical bonds for subsequent functionalization, following the Draslovka a.s., Czech Republic, non-public internal procedure. PAN fibers were then functionalized (NanoProgress, Pardubice-Polabiny, Czech Republic) by the covalent immobilization of anti-E. coli antibodies mentioned earlier by NanoProgress. For each of the experiment the functionalization was made immediately before the experiment, in order to eliminate the storage/decay effect. The final concentration of the bonded antibodies was determined by infrared (IR) spectroscopy (IRAFfinity-1; Shimadzu, Kyoto, Japan) at 1750 s−1  and was estimated to be 108 ± 12 µM g−1.

System for air filtration

A special pump system to carry out the experiments was constructed to detect bacteria in the air (Fig. 1, Scheme 1). The designed system consisted of a mechanical pump with a sealed chamber, a nebulizer (compressor inhaler InnoSpire ESSENCE; Philips, Amsterdam, Netherlands), and a container with bacteria. A system of pipes modified with metal filters enabled one-way airflow through the sealed chamber into which the fiber membranes were placed.

Figure 1 Air filtration system consisting of a mechanical pump, filter vessel, container for the sample, and nebulizer that maintained sufficient humidity for the nanobiosensors used.

Scheme 1 Scheme of the presented air filtration system.

Created by Biorender.com.

For most experiments, the nebulizer was connected to the pump system. Its role in the system was to maintain sufficient humidity for the antibody-functionalized fibers. Without suitable humidification, the antibody immobilized in the structure of fibers lost its bioactive properties.

The air filtration system was first tested without bacterial samples. To confirm the proper function of the system, filtration was performed by using fluorescein solution (the fluorescent xanthene dye dissolved in distilled water). After filtration of the fluorescent sample through the fiber membrane, the amount of captured color (intensity of the fluorescence) was measured by using a spectrofluorometer (Fluoromax-4; Horiba Scientific, Kyoto, Japan).

After testing and completion, the system was placed in a laminar box to ensure safety during the experiments.

Bacterial culture

Bacterial cultivation was performed on solid agar medium. The agar medium was prepared from 2.5 g of yeast extract, 2.5 g of peptone, 1.125 g of NaCl, 5 g of agar, and 250 ml of distilled water (Elbing & Brent, 2018). After heating and homogenization, the medium was poured into Petri dishes. Prepared Petri dishes with the medium were stored in a refrigerator at 4 °C.

From the reference strain, a single colony of E. coli was transferred to agar medium using the streak plate method. Passaged bacteria were cultured at 37 °C for 20 h (mini-incubator ICT 18; FALC Instruments, Lombardy, Italy) for each experiment (Noor et al., 2013).

Detection of bacteria

Every experiment took place for three days. The bacteria were passaged to new medium and then incubated for 20 h on the first day. On the second day, a single bacterial colony was transferred to the test tube with 15 ml of distilled water. The created bacterial suspension was stirred well. Three milliliters of the suspension were transferred into a cuvette, and the optical density of the suspension was measured with a spectrophotometer (λ = 600 nm, spectrophotometer UV-3600; Shimadzu). The OD600 were 0.152, 0.191, 0.193, 0.193 and 0.088 (which corresponds to 1.22 × 108, 1.53 × 108, 1.54 × 108, 1.54 × 108 and 7.04 × 107cells/mL) in case of the experiment determining the effectivity of the functionalization. In case of the experiment determining the influence of humidity, the OD600 were 0.152, 0.193, 0.072 and 0.096 (which corresponds to 1.22 × 108, 1.54 × 108, 5.76 × 107 and 7.68 × 107 cells/mL), respectively. The total binding capacity of the used fiber discs was 5 × 1012. The remaining 12 ml of the bacterial suspension was used to fill the nebulizer. A fine aerosol of bacteria in the air was created with the nebulizer. Meanwhile, the prepared fiber membrane was cut into approximately 1.5 × 1.5 cm pieces and washed with distilled water. The washed fiber piece was placed into the sealed chamber and used as a membrane for air filtration. For the detection of bacteria, the container was filled with bacterial aerosol. A mechanical pump filtered the entire container volume through the fiber membrane in the sealed chamber. After filtration, the fiber membrane was removed from the chamber and washed in distilled water using an ultrasonic cleaner (Geti GUC 601, 0.6 l). Eventually, the washed fibers were placed onto Petri dishes with agar medium and incubated at 37 ° C for 20 h. On the third day, Petri dishes were removed from the incubator, and the individual samples of the fiber membranes were transferred into test tubes with 10 ml of distilled water. The test tubes were stirred well, and the optical density of the bacterial suspensions was measured using a spectrophotometer.

Samples of unmodified PAN fibers and samples of functionalized PAN/Anti-E.coli fibers were prepared for each experiment.

In addition to the experiments during which the nebulizer was used, other experiments to prove the importance of humidity were conducted. The process of these experiments was almost identical to the procedure of the experiments explained above. To prove the importance of the proper environment for the antibody, the nebulizer was disconnected from the air filtration system. The bacterial culture was scattered into the container without added humidity.

Data analysis

The number of bacterial cells bonded to the functionalized antibodies was determined through optical density measurement. The OD600 of bacterial suspensions, which is the optical density at 600 nm, was measured using a UV-3600 spectrophotometer (Shimadzu).

The obtained data were compared to each other, and the resulting effectiveness of the prepared biochemical receptor was determined. The two tailed t-test was carried out using MS Excel built-in function, the significance level was set up to p < 0.05.

Results & Discussion

Functionalization and characterization of PAN fiber membranes

PAN fiber membranes were prepared by the electrospinning method and used as a biosensor matrix for bacterial detection in air. Due to their high surface-to-volume ratio, these membranes are incomparable materials for functionalization. The membranes were endowed with biosensing ability by specifically immobilizing an antibody onto their surface.

PAN was used to fabricate the electrospun fibers due to its great stability, insolubility in water, and good mechanical and chemical properties (Yardimci et al., 2022; Awad et al., 2021). PAN fibers were fabricated onto spun-bond fabric and characterized by scanning electron microscopy (SEM) (Fig. 2). As shown by the SEM analysis, the structure of the fabricated electrospun fibers was mostly regular without any significant heterogeneities. The mean diameter of the fibers ranged between 500 and 900 nm.

Figure 2 SEM images of unmodified PAN nanofibers at different magnifications.

After fabrication and characterization, the electrospun fibers were chemically modified and functionalized. These modifications were also characterized by SEM. The SEM analysis showed differences between the surface-modified (Fig. 3A) and antibody-functionalized PAN fibers (Fig. 3B). Compared to the unmodified fibers, the surface-modified fibers lost their smooth structure.

Figure 3 SEM images of electrospun PAN nanofibers.

SEM images of electrospun PAN fibers (A) with a chemically modified surface and (B) immobilized anti-E. coli antibodies, (C) with attached E. coli cells under dry conditions, (D) with attached E. coli cells under conditions of sufficient water availability (nebulizer).

The combination of chemicals used for fiber surface modification was significantly volatile, so the modification process was necessarily carried out with caution. In addition to this safety consideration, the delicate nature of the modification is also worthy of consideration. If this process is improperly performed, it can result in the destruction or structural degradation of fibers.

Unfunctionalized and functionalized fibers were used as membranes for the detection of the aforementioned airborne bacteria. As a membrane, the active diameter of the fibers was one cm. This diameter was determined by the grip used in the air filtration system.

Air filtration system

The detection of pathogenic organisms from the air using biosensors is very problematic. The biosensing layer (bioreceptor) of the biosensor is extremely sensitive to the environment. Whether they are enzymes or antibodies, the bioactive molecules used to detect pathogens require constant conditions, such as the pH and temperature, but mainly constant humidity. However, under atmospheric conditions, the humidity needed for the proper function of bioactive molecules is not sufficient. Therefore, to enable the detection of airborne bacteria using a biosensor, it was necessary to design a special air filtration system that could maintain constant environmental humidity. For the experiments, a mechanical pump was equipped with filters to ensure safety and a nebulizer to maintain proper humidity. Air moistened by the nebulizer was then sufficient for maintaining antibodies in their active form. In addition to the antibody activity, the intactness of the fibers was also maintained by the moistened air. Due to the functionalization process occurring in liquid media, the desiccation of functionalized fibers leads to their degradation.

Detection of specific markers of E. coli in air

The detection of E. coli was performed using the specifically designed pump system. However, to detect airborne bacteria, it was necessary to modify the system and incorporate an element that could disperse individual bacterial cells into the air. A nebulizer, which generated a fine mist containing bacteria, was used for this purpose. Another advantage of the nebulizer was its ability to moisten the functionalized fiber membranes. The constant humidity of the environment was critical for the preservation of the activity of the immobilized antibody. Without constant environmental conditions, the fiber-bound antibody could lose its effectiveness and specific biological activity. A 1.5 L container was filled with a fine mist to carry out experiments under a sufficient environment and used as a model environment. In experiments carried out to verify the importance of air humidification for the proper function of antibodies, bacterial cultures were scattered in a sample container without any added water.

Fiber membranes with an active diameter of one cm were used for air filtration and the detection of bacteria. Every experiment consisted of the filtration of three samples of unmodified PAN fibers and three samples of functionalized PAN fibers. After filtration, the fiber membranes were washed using an ultrasonic cleaner. Most bacterial cells become mechanically trapped in the porous fiber structure during filtration. Ultrasonication allows the removal of mechanically trapped cells, which is important for evaluating the effectiveness of antibody activity. Every fiber sample was cleansed individually so that the samples did not interfere with each other. After ultrasonic washing in distilled water, the fiber membranes were transferred onto Petri dishes in groups of three and then incubated overnight (Fig. 4). After 20 h of incubation, it was possible to observe bacterial colonies formed from the captured bacterial cells.

Figure 4 Petri dishes with PAN nanofibers.

(A) Unmodified PAN NFs directly after filtration; (B) unmodified PAN NFs after 20 h of incubation; (C) anti-E. coli PAN NFs directly after filtration; (D) anti-E. coli PAN NFs after a 20-hour incubation.

Measurement of the optical density

The number of bacteria was evaluated by measuring the optical density (OD600) of bacterial suspensions. Bacterial suspensions were prepared by transferring the fiber membranes into test tubes filled with distilled water. The suspensions needed to be sufficiently mixed for preparation so that the bacterial cells became released from the fiber structure. The samples were measured directly after their preparation and homogenization.

For the optical density measurement, 3 ml of each sample was transferred into clean cuvettes. The analysis was performed at a wavelength of 600 nm. This value was chosen because the molecules of biological materials do not absorb light of this wavelength, and thus, it is possible to eliminate the absorption of radiation by these molecules and directly measure the optical density of the prepared samples (Van Alst et al., 2023).

Individual samples of the bacterial cells captured by the functionalized and unmodified fibers were compared (Fig. 5). With the prepared functionalized PAN fibers, it was possible to increase efficiency up to 30%. However, the statistical significance between tests varied from p-value =0.014 (highly significant) to p-value =0.408 (non-significant), with the majority falling between p-value =0.05 to p-value =0.1, suggesting that more experiments have to be carried out. In addition to evaluating the functionalization effectiveness, the results from the measurements with and without a suitable environment were compared (Fig. 6), with similar significance levels as in previous experiment (p-values varied from 0.001 to 0.561). To compare the samples, it was necessary to carry out the measurements within a short period of time so that the nanofibrous membrane would not dry out. Functionalized fibers dried in air are susceptible to degradation due to their structural modifications. Given this fact, the connection of the nebulizer to the air filtration system is essential. The inability to provide necessary conditions for the bioactive layer also leads to its deactivation, thus decreasing detection efficiency. Functionalized fibers would thus lose their advantages, and their sensitivity would become comparable to that of unmodified fibers.

Figure 5 Comparison of the optical density (OD600) to the number of entrapped bacterial cells on the unmodified and functionalized nanofibers.

The asterisk (*) in the chart indicates the percentage difference in the effectiveness of the functionalized and unmodified nanofibers.

Figure 6 Comparison of the optical density (OD600) to the number of entrapped bacterial cells.

Differences among the functionalized nanofibers in humid air, air without sufficient humidity, and the unmodified nanofibers.

With the decreasing initial concentration of the bacterial suspension, the differences in the efficiency of the used nanofibrous membranes were significant (all p-values <0.05) (Fig. 7). Therefore, it promises future application of these membranes as ultrasensitive biosensors aiming mainly to very low analyte concentrations.

Figure 7 Comparison of the optical density (OD600) to the number of entrapped bacterial cells.

The differences in the number of captured bacteria are more noticeable with the lower initial concentration of bacterial suspension.

Conclusions

Due to their significant physical and mechanical properties and enormous surface-to-volume ratio, PAN fibers have garnered interest regarding sensor and biosensor application. Their specific structure also enables the possibility of additional functionalization. In this work, we present a proof of concept method for pathogen detection in air aiming to further development. To achieve this, an ultrasensitive fiber-based biosensor with immobilized antibodies specific to the detected pathogen has been fabricated. However, the direct detection of pathogens in air and air monitoring using biosensors are currently unachievable due to the lack of sufficient amount of water molecules necessary for the proper function of corresponding bioreceptors. In this work, this obstacle has been overcome by the deployment of our developed special air filtration system with a connected nebulizer maintaining the proper environment necessary for the biosensing layer.

The biosensor was tested using the model bacterial organism of E. coli. In addition to evaluating the functionalization effectiveness, the need to maintain sufficient humidity for air monitoring was estimated. Although the structure of fibers is so complex that they are suitable for air and liquid filtration by themselves, making the detection more difficult by trapping also the bacteria by other means than antigen-antibody reaction (Canalli Bortolassi et al., 2019), the immobilization of bioactive molecules can significantly increase the effectiveness of detection of up to 30%. Maintaining an environment suitable for bioreceptors also plays a significant role in the aforementioned increase in biosensor sensitivity. The loss of humidity can negatively affect the antibody activity and degrade the structure of functionalized fibers. Functionalized fibers in environments without a constant and sufficient humidity lose their selectivity and sensitivity for capturing and detecting biological particles, leaving the filtration properties only

The presented system demonstrates the possible way in overcoming the problems associated with insufficient water availability and thus enables the use of bioactive molecules in air sampling without further modification. The corresponding experiments demonstrated that maintaining properly humidified air significantly affected the amount of captured and thus detected bacterial cells. Moreover, the fabricated anti-E. coli PAN fibers used in combination with the presented air filtration system provided an easy, rapid, inexpensive, and especially sensitive solution for possible pathogen monitoring in air.

Supplemental Information

Supplemental Information 1 Raw data

The authors are grateful to their respective laboratories for providing lab space.

Additional Information and Declarations

Competing Interests

Author Contributions

Data Availability

The authors declare there are no competing interests.

Leontyna Varvarovska conceived and designed the experiments, performed the experiments, analyzed the data, prepared figures and/or tables, authored or reviewed drafts of the article, designed and built the sampling instrument, and approved the final draft.

Bruno Sopko conceived and designed the experiments, analyzed the data, prepared figures and/or tables, authored or reviewed drafts of the article, designed the sampling instrument, and approved the final draft.

Dana Gaskova performed the experiments, authored or reviewed drafts of the article, built the sampling instrument, and approved the final draft.

Tomas Bartl performed the experiments, authored or reviewed drafts of the article, built the sampling instrument, and approved the final draft.

Evzen Amler conceived and designed the experiments, authored or reviewed drafts of the article, and approved the final draft.

Tatana Jarosikova conceived and designed the experiments, authored or reviewed drafts of the article, and approved the final draft.

The following information was supplied regarding data availability:

The raw data are available in the Supplementary File.

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
