# Peer review of "Surface-functionalized PAN fiber membranes for the sensitive detection of airborne specific markers"

_PeerJ, doi:10.7717/peerj.18077_

## Round 0.1 · original submission · Major Revisions

Dear authors,

Thank you for submitting your work to PeerJ. The manuscript presents a relevant study on the application of nanofibers for the detection of airborne microorganisms and I think has a lot of potential to expand its applications. However, the three reviewers found issues and questions that need extensive and careful revisions. I kindly request to take the criticism positively and address them point by point. The manuscript will be revised once again by the experts. Please take sufficient time to response the reviewers.

I am looking forward for your revised manuscript.
All the best for all of you at the closure of 2023.
Bernardo

·

Basic reporting

The manuscript entitled Surface-Functionalized PAN nanofiber membranes for the sensitive detection through specific markers describes incorporating anti-Ecoli antibodies for specific airborne pathogen detection under air conditions. Despite the novelty proposal in designing polymeric sensors for air pathogens, the optical density of bacterial suspension between the untreated PAN and modified PAN is inconclusive. Also, the manuscript shows a poor discussion of the results and the comparison with other references. Check the following references:

DOI: 10.1002/adem.201700572
https://doi.org/10.1021/acsomega.0c02735
https://doi.org/10.3390/nano9121740
https://doi.org/10.1007/s00253-018-9051-0

An exhaustive literature review is required to expand the context and novelty of the research.

Experimental design

1. Introduction section (page 7, lines 76-82). The authors need to specify the antibody employed (include references associated with their use in the pullulans detection) and mention the antibody stability when incorporated into other polymer matrices.
2. Page 7, lines 83-87. Include the novelty of the work.
3. Materials & methods: Include the Mw of PAN, the viscosity of the solution, and the weight percentage of PAN in DMF employed for fabricating the fibers. Also, the electrospinning parameters (flow rate, voltage, distance, kind of collector, among others) must be described. Also, the membrane thickness must be mentioned.
4. Page 8, lines 115. Describe in detail the chemical modification-reduction reaction at the surface of PAN fibers.
5. Page 8, lines 117-121. The antibody functionalization at the surface of PAN fibers is unclear. Please explain in detail.
6. Bacterial culture section (page 8, lines 148-154) What is the initial bacterial concentration (µg/mL) employed for the detection assays (bacterial aerosol)? Is this bacterial concentration relevant for the antibody concentration in PAN fibers? This information must be clearly explained in the discussion section.
7. Functionalization and characterization of PAN membrane (page 0, lines 195-204). The explanation related to the surface functionalization by their enhanced surface area is weak and does not justify using PAN polymer for the sensor. Also, the average size of fibers is ca. 500-900 nm. Therefore, nanosized materials are of ca. 1-100 nm. Please modify the term “nanofibers” in all the manuscript (including the title) and change it to “fibers”.
8. Page 10, lines 208-209. “…the electrospun nanofibers were chemically modified and functionalized…” This could be unclear if it does not specify what kind of chemical modification was previously induced on the surface of PAN fibers. Please clarify.
9. Figs 2 and 3. SEM images of untreated PAN, chemically modified PAN, and antibody-functionalized PAN can be synthesized in only one figure (at the same magnifications) to compare changes in fiber diameter and surface defects produced by the surface treatment.
10. Discussion related to the PAN membrane as a bacterial sensor requires an extensive literature revision for comparison.
11. Figure 4. The petri dishes with PAN membranes untreated and treated with the antibody do not show the percentage of bacterial retention. Also, the quantitative differences in Fig 5 show slight differences in the bacterial growth with the surface modification. Therefore, the results are not conclusive.

Validity of the findings

It is required that the novelty of the manuscript be made explicit at the end of the introduction. As the results are inconclusive, please improve the conclusions.

·

Basic reporting

It is an interesting manuscript and well written, however it has major issues regarding the results and the data analysis, some experiments lack important controls, etc.

Experimental design

Figure 2: it would be better to also show the functionalized PAN fibers with trapped E. coli.
What is the selectivity of the method? Based on your results the PAN fibers by themselves, without functionalization are able also to trap E. coli, hence if the method is applied for more complex samples likely other kind of bacteria will be also trapped.
A Control experiment using other bacterial species alone and combined by E. coli is lacking.

Validity of the findings

Findings are not valid due a lack of statistical analysis, in particular for figures 5, 6 and 7 , differences are shown, some of them not very large, others seem significant but no statistical analysis was shown.
Without these analyses any finding cannot be validated.
Also in L 301 is stated “the differences in the efficiency of the used nanofibrous membranes were more significant (Fig. 7).” But not statistic test is shown!!!

Additional comments

In line 72 “Escherichia coli” is mentioned the first time, after that all times it should be abbreviated as “E. coli”.

L 94 “gram” Should be capitalized “Gram”.

Please provide more details about how the PAN fibers were functionalized.

Reviewer 3 ·

Basic reporting

The manuscript is well written. It is very clear and unambiguous.
The given context is quite clear and with enough literature references.
Manuscript structure is professional regarding structure, figures, tables, and supplemental material.
All results are appropriate for the designed detection system.

Experimental design

The manuscript fits well in the aims & scope (Biological Sciences, Environmental Sciences, Medical Sciences, and Health Sciences) since it deals with the detection of microorganisms that are harmful to human health.
It introduces a specifically designed detection system that can be used to overcome the problems associated with insufficient water availability and thus enables the use of bioactive molecules in a nanofiber sensor in air sampling without further modification.
The investigation was performed to a high-technical end. Materials and methods used are described clearly, such as nanofiber fabrication, functionalization of PAN fibers, a system for air filtration, bacterial culture, detection of bacteria, and data analysis by optical density measurement. They can be replicated.

Validity of the findings

The results are valid and replicable and are statistically sound.
Conclusions are well-stated and related to the original aim of the study and are limited to the obtained results.

Additional comments

This is a well-written manuscript that clearly defines the problem of detecting airborne microorganisms. In my opinion, it should be published as is.

---

## Round 0.2 · Major Revisions

Dear authors,

Please address all the concerns rise by the reviewers. The most critical issues should be taken into account and as suggested by Reviewer 1 please modify the title of the manuscript.

All the best for your research moving forward,

Bernardo

·

Basic reporting

The comments from the first review were appropriately resolved. However, due to the size of the fibers, I suggest modifying the title of the article and abstract (particularly the word nanofibers for fibers) to:
Surface-functionalized PAN fiber Membranes for the Sensitive Detection of Airborne Specific Markers

Experimental design

No comments

Validity of the findings

No comments

Additional comments

No comments

·

Basic reporting

no further comment.

Experimental design

Thanks for addressing my comments, however, as you mention more experiments need to be done in order to determine if the differences shown in figures 5 and 6 are significant.

Validity of the findings

Thanks for addressing my comments, however, as you mention more experiments need to be done in order to determine if the differences shown in figures 5 and 6 are significant.

Additional comments

no further comment.

---

## Round 0.3 · Minor Revisions

Dear authors,

Please revise the manuscript for grammar, spelling, typos and any grammar editing. Please address the comments regarding the statistical analysis used.

All the best,
Bernardo

·

Basic reporting

The previous revisions were properly attended. However, the manuscript requires an adequate review of typos, grammar, and spelling.

Experimental design

No comments

Validity of the findings

No comments

Additional comments

No comments

·

Basic reporting

Thanks for addressing my comments, make new experimental replicas and the statistical analysis, however, you also must:

1) Add a paragraph to materials and methods explaining how the statistical analysis was done (test used, program, and p-value considered significant).
2) Add the statistical significance to the figures (label statistical significant differences in the figure) and mention in the legends which differences were significant.

Experimental design

no comment

Validity of the findings

no comment

Additional comments

no comment

---

## Round 0.4 · accepted · Accept

Dear authors,

Thank you so much for addressing all the issues found by the reviewers. Congratulations on your paper, and thank you for choosing PeerJ for submitting your work.

Good luck with your research moving forward.

Warm regards,
Bernardo